# Trajectory representation learning for Multi-Task NMRDPs planning

## Abstract

Expanding Non Markovian Reward Decision Processes (NMRDP) into Markov Decision Processes (MDP) enables the use of state of the art Reinforcement Learning (RL) techniques to identify optimal policies. In this paper an approach to exploring NMRDPs and expanding them into MDPs, without the prior knowledge of the reward structure, is proposed. The non Markovianity of the reward function is disentangled under the assumption that sets of similar and dissimilar trajectory batches can be sampled. More precisely, within the same batch, measuring the similarity between any couple of trajectories is permitted, although comparing trajectories from different batches is not possible. A modified version of the triplet loss is optimised to construct a representation of the trajectories under which rewards become Markovian.

## 1 Introduction

One of the major objectives in reinforcement learning is to identify optimal policies of a given Markov Decision Process (MDP), whose existence is typically ensured if the environment is Markovian (Sutton et al., 1998). However, this assumption is usually not fulfilled in many real life problems. A first possible cause is "partial observability", when the real state space is observable only through a projection on a sub-space. For instance, consider a (cleaning) robot that uses the distance from the walls in four directions as a state space. This representation of the environment does not necessarily correspond to the exact location in the house as different places share the exact same distances to walls. This setting is usually framed as a Partially Observed Markov Decision Process (POMDP) (Shani et al., 2013; Cassandra, 1998; Murphy, 2000; Hausknecht & Stone, 2015; Zhu et al., 2018). A second possible cause is that reward functions might depend on the whole trajectory. Consider the example of a waiter/server robot whose state space is the current position in the coffee shop and the current interaction with the clients (taking orders, getting and delivering beverages). The objective is to train this robot to bring coffee when ordered. A possible reinforcement signal is to reward the agent when it processes an order that was not satisfied in the past. As the states do not include past requests, nor their fulfilment, the environment is not Markovian and the proposed reward is a function of the trajectory up to the present. This setting is a Non-Markovian Reward Decision Process (NMRDP) (Bacchus et al., 1997; Thiébaux et al., 2006; Bakker, 2002; Bacchus et al., 1996). Both POMDP and NMRDP are special cases of Non Markovian Decision Processes.

The second above example (waiter robot) is actually a multiple task problem. Planning in this setting is a challenging problem because it requires more than optimally and independently solving each task (Toro Icarte et al., 2018). It is possible that a global optimal policy for the whole sequence of tasks induces sub-optimal policies for each one of the sub-tasks. Let us take a naive example of an agent building a shelter which requires gathering wood and stones from nearby resources. To gather wood/stones optimally, the agent should look for the nearest forest/quarry. However, if it wants to optimally gather all the resources, it should not just consider how close the forests or the quarries are, but how close they are to each other as it will have to walk from one to the other. Notice that this example shares similarities with the k-server problem (Manasse et al., 1990).

In those cases, the nature of the task is not encoded in the state space, so the Markov assumption is not verified. Casting the problems as an NMRDP requires using a (stage by stage) reward function that encodes the tasks specifications. In the shelter building scenario, the reward as a function of the states, while none of the tasks is achieved, is different from when stones or wood have been collected. If it is possible to map the previously observed states (the trajectory up to the present)

into the current task, the problem can be expanded into a standard MDP, where it is known how to identify optimal policies. Motivated by the shared similarities of the described problems, we shall focus on learning to identify current tasks from the trajectory of visited states (Wilson et al., 2007). Ideally, if annotated trajectory data sets are available, learning a trajectory representation function that expands the NMRDP into an MDP would be a supervised task. However this is a strong assumption. Indeed if a simple way of creating such data sets existed, it would be used to expand the NMRDP's state space. Another issue, is that given a set of tasks, there might exist multiple non Markovian reward functions of interest. For instance, consider an agent whose goal is to maximise customers satisfaction. In this case the reinforcement signals are the declared satisfaction of said customers (in the form of a rating for example). However even if different customers expect the same goals to be accomplished by the robot, their perception of the performance might differ.

This example can be generalised to the case where several experts (or simply users) are training a robot to perform some tasks, possibly different from one expert to the other. They share a common NMRDP environment with different reward/reinforcement functions as the set of tasks (or their appreciations/preferences) might be different from one expert to the other. The same augmented MDP can be used for all of their respective NMRDP as they share the same latent global set of tasks (more or less whatever the robot can physically do). This training can be made by the expert independently but it will certainly be faster and more efficient if made collectively. In these cases, in order to identify the full set of tasks and annotate the trajectories, a central authority must be aware of the reward function of each expert. This might be impossible due to tractability reasons, or privacy constraints. Fortunately, the experts can easily provide "similarity" between trajectories with respect to the latent task, i.e., whether the robot was performing the same task or not at that time, without specifying precisely the task. In the customer example, the evaluations given by the same client can be used to compare if two trajectories are associated with the same task or not, but not if those trajectories come from batches associated to two different customers. Indeed, establishing comparisons between the tasks associated to subsets of the observed trajectories without actually knowing the actual task is possible. As a consequence, we are going to assume that learning is done using batches of trajectories, where we will have a task similarity measure within each batch.

In order to identify the tasks, standard techniques identify a latent state space in which policies become Markovian, thus optimal ones exist. Unfortunately these approaches require domain specific knowledge in the form of "relevant propositions" (Bacchus et al., 1997; Thiébaux et al., 2006; Bakker, 2002; Toro Icarte et al., 2018), either by estimating the hidden state or by constructing additional features. For example, in the waiter robot case, received orders are the relevant propositions that the agent needs to save in order to identify current tasks. Our major contribution in this paper is a generic approach to discover a latent state space given a set of similar and different trajectories. This assumption requires weaker domain specific knowledge.

Once the latent state space is available, multi-task and meta-learning related approaches provide efficient tools to identify optimal policies (Andrychowicz et al., 2017; Colas et al., 2019) or to learn a general purpose policy from which learning a specific task's optimal policy can be done in few steps (Duan et al., 2017; Rakelly et al., 2019).

The remaining of the paper is organised as follows: we introduce the mathematical tools used in this paper and we explain how they relate to our work in Section 2. We specify in Section 3 a subset of NMRDPs that can be expanded with a Markovian latent space into MDPs and we provide a possible structure of such expansion. In Section 4, we propose an algorithm to learn trajectory representation functions that approximate the tasks' latent state space and thus approximates the proposed equivalent MDPs. Section 5 is dedicated to empirical results. We consider toy NMRDPs grid-world environment (where we solve a multi-task problem using learned representations) and a real-life data set of tourists visiting Salzburg (where we learn a trajectory representations that clusters their paths according to their destinations).

## 2 BACKGROUND

### 2.1 NON MARKOVIAN REWARD DECISION PROCESS (NMRDP)

Both finite NMRDPs and MDPs are defined by a finite state space $\mathcal{S}$, a finite action space $\mathcal{A}$, a transition probability $\mathcal{P}$, a discount factor $\gamma$, and a reward function $\mathcal{R}$. In NMRDPs, the reward function maps trajectories into real values while in MDPs the reward function maps states into real

values. To the extent of our knowledge, existing methods for planning in NMRDP (Toro Icarte et al., 2018; Bacchus et al., 1997; Thiébaux et al., 2006) define an augmented state space to build an equivalent MDP to the NMRDP. The most straightforward idea would be to define the augmented state space $\mathcal{S}^*$ as the set of all possible trajectories. However, the number of possible trajectories grows exponentially with the horizon, which makes the numerical complexity intractable for state of the art reinforcement learning techniques.

An established approach to expand the state space requires a set of domain-specific propositions. Using temporal logical operator it combines these propositions to automatically construct a set of temporal variables (Bacchus et al., 1997; Thiébaux et al., 2006) that extends the state space to specify an equivalent MDP. These methods have been successfully used to teach a reinforcement learning (RL) agent multiple tasks (Toro Icarte et al., 2018). However, the downside is that domain specific propositions are required to disentangle the non Markovianity of the reward function. Other approaches exploit the model free RL techniques by adding a Long Short Term Memory (LSTM) layer to the policy network (Bakker, 2002). The agent policy becomes a function of the full trajectory and the agent has enough information to find the optimal policy. In the tested toy examples of POMDPs, the agent needs undesirable directed exploration techniques to have a good performance (Bakker, 2002). Moreover, when dealing with similar NMRDPs with different reward functions, we need to start the training from scratch when using this approach.

Motivated by the lack of approaches to expand the state space when relevant information about the reward structure are not provided, we investigate the state expansion problem in NMRDPs.

## 2.2 TRIPLET LOSS AND CONTRASTIVE LEARNING

Learning useful representations of the data has been done in an array of challenging tasks. The access to data in the form of similar and/or dissimilar observations enabled the development of multiple efficient contrastive learning techniques (Mikolov et al., 2013; Schroff et al., 2015; Kilian & Lawrence, 2009). Recent work (Arora et al., 2019) establishes a link between contrastive learning and efficient linear separability of the data. The core idea is to find a representation in a metric space so that similar samples have nearby representations and vice versa.

The triplet loss is a clever evaluation of this idea. Multiple formulations have been suggested in the literature for both the loss definition and the batching procedure (Kilian & Lawrence, 2009; Arora et al., 2019; Ding et al., 2015; Oh Song et al., 2016; Hermans et al., 2017). A recent success of this approach lies in a new formulation (Hermans et al., 2017) that outperforms existing techniques in the person re-identification task (i.e., representing pictures of the same person from different angles as similar embedding vectors, and representing pictures of different persons as different embedding vectors). The suggested batching (called $PK$ Batch-hard) selects $P$ classes (person identities) and for each of these, $K$ examples (images from different angles). Each of the $PK$ images is selected as an anchor, and each of the anchors is associated with the hardest positive (the same person's image from an angle which representation is the furthest to the anchor's representation) and with the hardest negative (a different person's image which representation is the closest to the anchor's). This allows to build a batch of anchors, positives and negatives of size $PK$.

Formally, let $X$ be the set of the $PK$ sampled elements, with $X_i^j$ being the $i^{th}$ sample of the $j^{th}$ class, and $f_\theta(X_i^j)$ be a representation parameterised by $\theta$ in a metric space $\mathbb{R}^d$ where $d$ is the representation dimension. The metric of the representation space is denoted by $D(u,v)$. The proposed loss is:

$$\mathcal{L}_{BH}(\theta, X) = \sum_{j=1}^{P} \sum_{i=1}^{K} \left[ m + \max_{p \leq K} D(f_\theta(X_i^j), f_\theta(X_p^j)) - \min_{p \leq K, c \leq P, c \neq j} D(f_\theta(X_i^j), f_\theta(X_p^c)) \right]_+ . \quad (1)$$

The margin $m$ is a constraint on how far the negative samples should be from each other, and the relu function $[x]_+$ avoids correcting already well defined representations. The relu function can be substituted by any smooth approximation such as $\log1p(.)$ defined as $\log1p(x) = \log(1 + e^x)$.

In multi-task problems, a representation that identifies the current task is sufficient to expand the state space and build the equivalent MDP. Given trajectory batches with similar and dissimilar tasks, the contrastive learning approaches, and particularly the triplet loss, lend themselves to construct

trajectory representations that translate the task similarity. In these approaches, no information about the reward structure is required.

## 3 LATENT SPACE FOR MULTI-TASK NMRDPS

Given an NMRDP denoted by $\mathcal{N} = \{\mathcal{O}, \mathcal{A}, \mathcal{P}, \mathcal{R}, \gamma\}$ where $\mathcal{O}$ (for "observable") is the state space, and the reward is a function of the trajectory $(o_t)_{t=0}^{t=T}$, there always exists an equivalent MDP that extends $\mathcal{N}$ (Bacchus et al., 1996).

Consider multi-task problems that do not suffer from partial observability. In other words, if the NMRDP $\mathcal{N}$ is a formulation of the multi-task problem, then the reward function is Markovian in the observables and some hidden goals (it depends on what task is being achieved and what current state $o_t$ is being visited). This restriction implies that given the nature of the current task, the agent is able to make optimal decisions based on the current observation. Many real life problems (like the examples described in the introduction) satisfy this constraint naturally.

Recall that in an NMRDP, the reward is a function of the whole trajectory $\mathcal{R}((o_t)_{t=0}^{t=T})$. However we assume that given the mapping $\mathcal{T}$ from trajectories into the set of tasks $\mathcal{H}$,

$$\mathcal{T} : \{((o_t)_{t=0}^{t=T}) \ \forall \ T \in \mathbb{N}\} := \Gamma(\mathcal{O}) \to \mathcal{H},$$

the reward function $\mathcal{R}$ can be rewritten as:

$$\mathcal{R}((o_t)_{t=0}^{t=T}) = \mathcal{R}(o_T, \mathcal{T}((o_t)_{t=0}^{t=T})) = \mathcal{R}(o_T, h_T) \quad \forall \ T \in \mathbb{N}. \tag{2}$$

In multi-task problems, the mapping $\mathcal{T}$ can be defined recursively using the recent observations and tasks instead of using the whole trajectory. Consider the shelter building scenario, where the tasks can be defined as "collecting the wood", "collecting the stones" and "building the shelter". If the agents is currently "collecting the stones", and given that it already "collected the wood", it only need to check the current state (to verify if it is currently in the quarry or not) to decide if we keep "collecting the stones" or go on to "building the shelter".

In these cases and depending on the nature of the problem, $\mathcal{T}$ is a bounded memory process in $\mathcal{O}$ and $\mathcal{H}$. Thus there exists a task horizon $\tau_T \geq 1$ and an observation horizon $\tau_O \geq 0$ such that:

$$\mathcal{T}((o_t)_{t=0}^{t=T}) = \mathcal{T}((o_{T-\tau})_{\tau=0}^{\tau=\tau_O}, (h_{T-\tau})_{\tau=1}^{\tau=\tau_T}) \quad \forall \ T \in \mathbb{N}.$$

To avoid cumbersome notations, we will restrict ourselves to the case where $\tau_O = 0$ and $\tau_T = 1$, however, our results can be generalised to any values of $\tau_O$ and $\tau_T$. In this case, $\mathcal{T}$ can be further simplified in the following form:

$$\mathcal{T}((o_t)_{t=0}^{t=T}) = \mathcal{T}(o_T, h_{T-1}) = h_T \quad \forall \ T \in \mathbb{N}. \tag{3}$$

In other terms, the considered NMRDPs admit an equivalent MDP formulation $\mathcal{M}^* = \{\mathcal{S}^*, \mathcal{A}^*, \mathcal{P}^*, \mathcal{R}^*, \gamma\}$ whose state space is $\mathcal{S}^* = \mathcal{H} \times \mathcal{O}$. By construction of the equivalent MDP (Bacchus et al., 1996), and under the consideration of Equation 3, $\mathcal{P}^*$ verifies the following:

$$\mathcal{P}^*((o_{t+1}, h_{t+1})|(o_t, h_t), a_t) = \mathcal{P}(o_{t+1}|o_t, a_t) \times \mathbb{1}_{h_{t+1} = \mathcal{T}(h_t, o_{t+1})}.$$

In other terms, if the MDP satisfies the following Equation 4, then $\mathcal{M}^*$ is equivalent to $\mathcal{N}$ (Bacchus et al., 1996):

$$\begin{cases} \mathcal{A}^*(o_t, h_t) & = & \mathcal{A}(o_t) \\ \mathcal{R}^*(o_t, h_t) & = & \mathcal{R}(o_{0:t}) \\ \mathcal{P}^*((o_{t+1}, h_{t+1})|(o_t, h_t), a_t) & = & \mathcal{P}(o_{t+1}|o_t, a_t) \times \mathbb{1}_{h_{t+1} = \mathcal{T}(h_t, o_{t+1})} \end{cases} \tag{4}$$

As $\mathcal{N}$ satisfies Equation 2, the construction of $\mathcal{M}^*$ is straightforward once the mapping $\mathcal{T}$ is known.

### 3.1 FEATURE REPRESENTATION FOR THE LATENT SPACE

Typical techniques to learn optimal policies in problems with large state space exploit feature functions (Lillicrap et al., 2015; Mnih et al., 2015; 2013). This motivated learning an embedding

$\phi : \Gamma(\mathcal{O}) \to \mathbb{R}^d$ where $d$ is the dimension of the tasks' feature space. As the goal is learning optimal policies using classical RL techniques, it is important to evaluate an MDP $\hat{\mathcal{M}} = \{\hat{\mathcal{S}}, \hat{\mathcal{A}}, \hat{\mathcal{P}}, \hat{\mathcal{R}}, \gamma\}$, equivalent to $\mathcal{N}$, such that $\hat{\mathcal{S}} = \mathbb{R}^d \times \mathcal{O} \supseteq \phi(\Gamma(\mathcal{O})) \times \mathcal{O}$. In what follows we construct the MDP $\hat{\mathcal{M}}$ as a reformulation of $\mathcal{M}^*$ by substituting $h_t = \mathcal{T}(o_{0:t})$ with $\phi(o_{0:t})$ in Equation 3 and Equation 4. We denote $\phi_t = \phi(o_{0:t})$.

Equation 4 implies that the feature-based equivalent MDP must satisfy the following properties:

$$\begin{cases} \hat{\mathcal{A}}(o_t, \phi_t) & = & \mathcal{A}(o_t) \\ \hat{\mathcal{R}}(o_t, \phi_t) & = & \mathcal{R}(o_{0:t}) \\ \hat{\mathcal{P}}((o_{t+1}, \phi_{t+1})|(o_t, \phi_t), a_t) & = & \mathcal{P}(o_{t+1}|o_t, a_t) \times 1_{\phi_{t+1} = \phi(o_{0:t+1})} \end{cases} \quad (5)$$

As a consequence, the construction of $\hat{\mathcal{A}}$ is straightforward from $\mathcal{A}$. The reward ($\hat{\mathcal{R}}$) and the transitions ($\hat{\mathcal{P}}$) are constructed based on Equation 4 and Equation 5 given any function $\phi$ that separate the tasks, i.e. there exist a classifier $\mathcal{C} : \mathbb{R}^d \to \mathcal{H}$ such that $\mathcal{C} \circ \phi = \mathcal{T}$:

$$\begin{cases} \hat{\mathcal{R}}(o_t, \phi_t) = \mathcal{R}^*(o_t, \mathcal{C}(\phi_t)) = \mathcal{R}^*(o_t, h_t) \\ \hat{\mathcal{P}}((o_{t+1}, \phi_{t+1})|(o_t, \phi_t), a_t) = \mathcal{P}^*((o_{t+1}, \mathcal{C}(\phi_{t+1}))|(o_t, \mathcal{C}(\phi_t)), a_t) \end{cases}$$

So from a practical point of view, the construction of an equivalent MDP to $\mathcal{N}$ boils down into approximating any trajectory representation function $\phi : \Gamma(\mathcal{O}) \to \mathbb{R}^d$, as long as $\phi$ separates the tasks.

If an agent is learning the optimal policy of $\mathcal{N}$ using any RL (Reinforcement Learning) setting, it will receive both the rewards $r_t = \mathcal{R}((o_t)_{t=0}^{t=T})$ and the observations $o_t$ over time. But as the problem is non Markovian, the optimal policy can not be expressed using solely the observations $o_t$. However if the same agent uses a trajectory representation function $\phi$, it can construct over time the latent task feature $\phi_t$. Using RL techniques, finding an optimal policy $\pi^*$ in the equivalent MDP $\hat{\mathcal{M}}$ (with the extended state space $(o_t, \phi_t)$) is possible, and by extension (Bacchus et al., 1996), $\pi^*$ is an optimal policy of $\mathcal{N}$.

## 4 APPROXIMATION OF A TRAJECTORY REPRESENTATION FUNCTION

Without loss of generality, consider the case where the NMRDP admits $n$ possible tasks: $\mathcal{H} = \{1, 2, .., n\}$. Let $\phi : \Gamma(\mathcal{O}) \to \mathbb{R}^d$ be a trajectory representation function that separates tasks such that:

$$\inf_{\gamma_1, \gamma_2, \mathcal{T}(\gamma_1) \neq \mathcal{T}(\gamma_2)} |\phi(\gamma_1) - \phi(\gamma_2)| > \sup_{\gamma_1, \gamma_2, \mathcal{T}(\gamma_1) = \mathcal{T}(\gamma_2)} |\phi(\gamma_1) - \phi(\gamma_2)|. \quad (6)$$

Let $e_i = \arg\min_{e \in \mathbb{R}^d} \max_{\gamma \ s.t. \mathcal{T}(\gamma) = i} |\phi(\gamma) - e_i|$ the centroid representations of each task. For a given trajectory representation $e \in \mathbb{R}^d$, the classifier $\mathcal{C}^*(e) = \arg\min_{i \in \mathcal{H}} |e - e_i|$ satisfies the following:

$$\mathcal{C}^* \circ \phi(\gamma) = \mathcal{T}(\gamma) \quad \forall \gamma \in \Gamma(\mathcal{O}). \quad (7)$$

Such classifier can be approximated in an unsupervised manner using K-means algorithm (Kanungo et al., 2002). Let $\hat{\mathcal{C}}$ be such approximation. As a consequence, Equation 7 simplifies the problem into finding any trajectory representation function $f$ that satisfies Equation 6 for any two trajectories $\gamma_1, \gamma_2 \in \Gamma(\mathcal{O})$. We propose in the remaining to approximate such function using deep neural networks as a parametric family $f_\theta$ and optimising the triplet loss introduced in Equation 1 with respect to $\theta$.

The obtained trajectory representation functions $f_{\theta^*}$ can either be used to reconstruct an approximation of $\mathcal{M}^*$ (by evaluating $\hat{\mathcal{C}} \circ f_{\theta^*}$) or to build an approximation of $\hat{\mathcal{M}}$ (by using $f_{\theta^*}$ as a feature function instead of $\phi$).

## 4.1 ARCHITECTURE OF THE TRAJECTORY REPRESENTATION FUNCTION

We have tested multiple architectures for the $f_\theta$ function. The most promising architectures were mostly inspired by the Long Short Term Memory (LSTM) cell architecture (Hochreiter & Schmidhuber, 1997) and the Gated Recurrent Unit (GRU) cell architecture (Cho et al., 2014).
The architecture that we use in our implementation builds features from the observed states using a Multi-Layer Perceptron (MLP) architecture (Rumelhart et al., 1985). The obtained features are concatenated with the last latent state $\phi_{past}$, and fed to two parallel MLPs. The first one generates a proposed latent state $\tilde{\phi}_{new}$, and the second generates a forget gate variable $Z \in [0, 1]$ (Hochreiter & Schmidhuber, 1997; Cho et al., 2014). The evaluated new latent state $\phi_{new}$ is computed as follows:

$$\phi_{new} = Z \times \tilde{\phi}_{new} + (1 - Z) \times \phi_{past}$$

## 4.2 OPTIMISATION OF THE TRAJECTORY REPRESENTATION FUNCTION

Batches of comparable trajectories are used in order to learn a representation function $f_\theta$ that separates tasks. To construct these batches in practice, the only domain-knowledge required is the ability to compare tasks. As mentioned before, the batches can eventually be collected from different experts working with different reward functions that share the same latent task space $\mathcal{H}$.
Let $(\gamma_i^j)_{i \leq K, j \leq P} \in \Gamma(\mathcal{O})$ be such that $\mathcal{T}(\gamma_{i_1}^j) = \mathcal{T}(\gamma_{i_2}^j)$. Given a representation function $f_\theta$, the representation $\phi(\theta)_i^j \in \mathbb{R}^d$ of the trajectory $\gamma_i^j$ is constructed recursively with Equation 3:

$$\phi_t = f_\theta(o_t, \phi_{t-1}). \tag{8}$$

In order to satisfy Equation 6, the batch hard loss function from Equation 1 is optimised with respect to $\theta$:

$$\mathcal{L}_{BH}(\theta, \gamma) = \sum_{j=1}^{P} \sum_{i=1}^{K} \log1\mathrm{p} \left( m + \max_{p \leq K} ||\phi(\theta)_i^j - \phi(\theta)_p^j|| - \min_{p \leq K, c \leq P, c \neq j} ||\phi(\theta)_i^j - \phi(\theta)_p^c|| \right) \tag{9}$$

Empirically, given the same computational power, using a special case of truncated back-propagation through time (Puskorius & Feldkamp, 1994) (where we update the weights one time step at the time after computing the representation of the whole trajectory) provided better performances. This approach can be seen as an EM (Expectation-Maximisation) algorithm. In the expectation step (E), the representation of the available data is evaluated under a fixed $\theta$ (i.e. $\phi(\theta)_i^j$ is computed). In the maximisation step (M), the recursive definition of task features in Equation 8 is used to define Equation 10, a local version of Equation 9:

$$\mathcal{L}_{BH}^{local}(\theta, \gamma, \theta_{old}) = \sum_{j=1}^{P} \sum_{i=1}^{K} \log1\mathrm{p} \left( m + \mathcal{L}_{\max}(\gamma_i^j, \theta, \theta_{old}) - \mathcal{L}_{\min}(\gamma_i^j, \theta, \theta_{old}) \right) \tag{10}$$

Let $o_i^j$ be the last observation in the trajectory $\gamma_i^j$, and let $\tilde{\phi}(\theta)_i^j$ the representation of the trajectory $\gamma_i^j$ without the last observation. We define $\mathcal{L}_{\max}(\gamma_i^j, \theta, \theta_{old})$ and $\mathcal{L}_{\min}(\gamma_i^j, \theta, \theta_{old})$ as follows:

$$\begin{cases} \mathcal{L}_{\max}(\gamma_i^j, \theta, \theta_{old}) &= \max_{p \leq K} ||f_\theta(o_i^j, \tilde{\phi}(\theta_{old})_i^j) - f_\theta(o_p^j, \tilde{\phi}(\theta_{old})_p^j)|| \\ \mathcal{L}_{\min}(\gamma_i^j, \theta, \theta_{old}) &= \min_{p \leq K, c \leq P, c \neq j} ||f_\theta(o_i^j, \tilde{\phi}(\theta_{old})_i^j) - f_\theta(o_p^c, \tilde{\phi}(\theta_{old})_p^c)|| \end{cases}.$$

Given a step size $\alpha$, the maximisation step (M) is a gradient descent with respect to $\theta$:

$$\theta_{new} = \theta_{old} - \alpha \nabla_\theta \mathcal{L}_{BH}^{local}(\theta, \gamma, \theta_{old}).$$

Basically, optimising Equation 10 in the M-step, assumes that the previous representations $\tilde{\phi}(\theta)_i^j$ is an input data and does not propagate the gradient further than the current observation. The E-step, re-evaluates the representations of the trajectory over time. By alternating these steps, we were able to obtain the provided experimental results.

## 5 EXPERIMENTS

As described before, building an equivalent MDP to an NMRDP $\mathcal{N}$ can be reduced to expanding the state space with a trajectory representation that separates tasks. Experimentally, we establish the

ability to approximate such function in a simulated toy example and in a real life problem.

We used an NMRDP Grid-world problem where the agent needs to visit a sequence of particular cells in order to be rewarded. The reward function is not Markovian because the state space is just the coordinates of the current cell. Using $PK$ batches from randomly generated trajectories, we construct a representation function that separate the tasks and thus expands successfully the state space. This is detailed in Section 5.1

For the real life example, we used the GPS tracks of tourists in Salzburg City, Austria. The tourists can be seen as agents who are rewarded from certain "attractions" the first time they see them. So this can be stated as an NMRDP problem. We build a representation function of the GPS tracks that clusters trajectories going to the same place and separate those heading to different directions in Section 5.2.

We also provide a github repository with our implementation of the solution (using *TensorFlow*) along with the necessary modules to simulate the toy example and the data of the real life example. In order to recreate the experimental results provided in what remains, you can access the contents on the following ***Microsoft Azure blob*** (This link is anonymous and respects the double-blind reviewing process, it will be substituted with a Github repository in the camera ready version)

## 5.1 GRID-WORLD PROBLEM

We run our approach on a Grid-World problems with varying grid dimensions. The state space $\mathcal{O}$ is the grid cells, the action space $\mathcal{A}$ is the available directions (North, South, East, West) and staying still. Transition probabilities $\mathcal{P}$ are naturally defined by the action space. We will consider a non Markovian reward, with latent state structure $\mathcal{H} = \{h_i, i \in [0, n]\}$. Each of the latent states $h_i$ is associated with visiting a particular "target cell" $\mathbf{T}(h_i)$. If the target cell of $h_i$ is visited, $\mathbf{N}(h_i)$ defines the next task/latent state. Equation 11 formally defines this setting.

$$\begin{cases} \mathbf{T}(h_i) = o_i \in \mathcal{O} \\ \mathbf{N}(h_i) = h_j \in \mathcal{H} \\ \mathcal{T}((o_t)_{t=0}^{t=T}) = \mathcal{T}(o_T, h_{T-1}) = \begin{cases} \mathbf{N}(h_{T-1}) & if \quad o_T = \mathbf{T}(h_{T-1}) \\ h_{T-1} & if \quad o_T \neq \mathbf{T}(h_{T-1}) \end{cases} \\ \mathcal{R}((o_t)_{t=0}^{t=T}) = e^{-|o_T - \mathbf{T}(h_T)|} \end{cases} \quad (11)$$

To simulate trajectories, we use a random policy to generate the action at each time step as this ensures generating a wide variety of trajectory patterns. We exploit the hidden states to construct the $PK$ batches. $P$ hidden states are sampled, and for each of them $K$ trajectories are identified to build the batch. Training uses the EM algorithm in order to optimise the representation function.

In order to evaluate a given representation function in this toy example, we plot, for a set of trajectories, the obtained representations $\phi_t \in \mathbb{R}^d$ and we colour label the points with respect to the hidden states $h_t \in \mathcal{H}$.

Figure 1 illustrates the results with 5 latent tasks, in a $5 \times 5$ grid. The graphs from left to right represent the following:

- A projection on the first two dimensions of the trajectories representations
- An unsupervised dimension reduction using PCA (Principal Component Analysis)
- A supervised dimension reduction using LDA (Linear Discriminant Analysis)

The obtained representations, when reduced to two dimensions, show clearly that trajectories have been separated according to the associated task. Even when using unsupervised dimension reduction, we are able to spot clusters of representations that can be matched with different tasks. The third plot (with LDA reduction) proves that the data in the original representation space ($\mathbb{R}^d$) can be separated linearly with respect to the tasks. We conclude that the construction of an n-class classifier (5 classes in the illustrated example) is possible. Using Equation 7, the function $f_\theta$ that satisfies Equation 6 and a classifier, we construct an approximation of a task feature function $\phi \circ \mathcal{T}$. Therefore, an approximation of the equivalent MDP $\mathcal{M}^*$ that expands the NMRDP.

When optimising the trajectory representation function, an annealing procedure increased the convergence speed and the quality of the results. A relaxed version of the batch hard triplet loss defined in Equation 9 is used. This allows exploring easier batches in the early stages of learning and generating harder batches as the representation gets better at separating the tasks. This is discussed further in Appendix A.

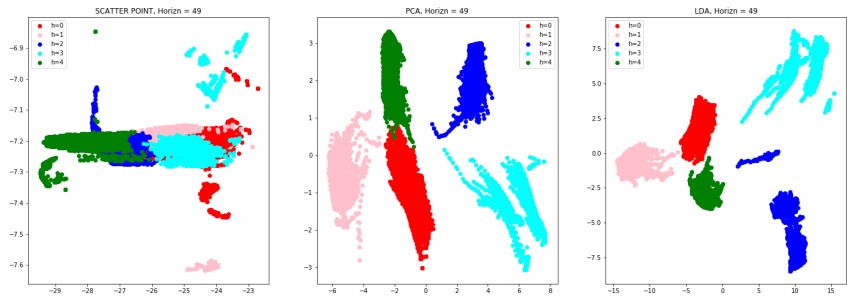

Figure 1: Evaluating the trajectory representation

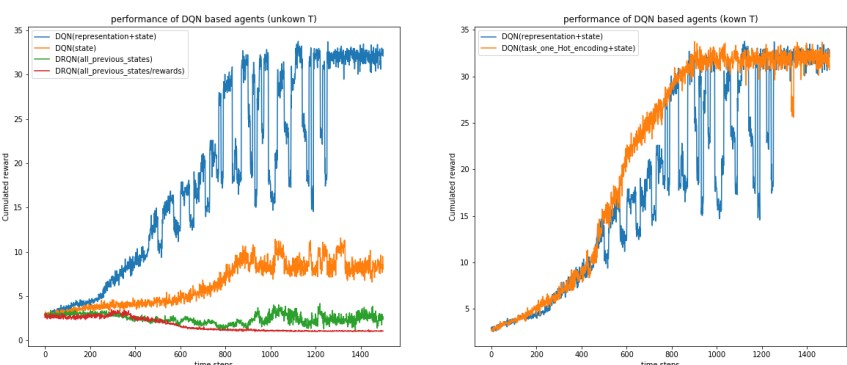

Figure 2: DQN performances with and without using the trajectory representation

Deep Q-Networks (Mnih et al., 2013) are one of the widely used off-the-shelf approaches in RL. The algorithm samples trajectories using an $\epsilon$ greedy policy with respect to the Q-function estimation and optimises a loss function based on the Bellman equation. By using a different neural network in the sampling steps than the one used for computing the parameter gradients, storing the recent trajectories in a buffer replay memory, and annealing $\epsilon$, the neural network converges to an unbiased estimator of the optimal Q-function and thus identifies an optimal policy. In order to establish that the obtained representation is valuable to learning the optimal policy, we compare the performances of DQN-based agents with and without the use of the representation, and with the use a one-hot-encoding of the actual task. The graph on the right in Figure 2 provides a comparison of the DQN performances when using only the state features, the state features along with the learned trajectory representation, and a recurrent version of the DQN (DRQN) (Hausknecht & Stone, 2015) provided the history of state features and rewards. Both the learned representations and the LSTM within the DRQN embed the trajectories in a latent space with the same dimension. In all experiments we used the same horizon, $\epsilon$ annealing step and the same replay memory size. As expected, using the learned representations enables the agent to outperform the settings where only the state features/rewards where provided. This represents quantitative proof that the evaluated trajectory representations are more useful to an agent learning downstream tasks than exploring previous rewards or previous observations within the policy network. The graph on the left side of Figure 2, illustrate a comparison between the performances of a DQN-based agent when using either the learned representations or a one hot encoding of the actual task. Both performances are comparable, confirming that the obtained representations expended the NMRDP into an MDP. However, using the one-hot encoding of the actual task enables the agent to converge faster. This is a consequence of using an approximation of the actual equivalent MDP. However with this comparison we establish that in such NMRDP scenarios, having access to the task function is not a requirement. The ability to identify similar/different trajectories with respect to the latent task is sufficient to converge to an optimal policy.

## 5.2 SALZBURG CITY: TOURISTS GPS TRACKS

The GPS tracking took place in Salzburg City, Austria. The participant of the data collection were tourists (44) residing in a youth hostel next to the old town (Kellner & Egger, 2016). Given that the collected trajectories reflect the behaviour of a first time visitor of the city, it is likely that there exist a common latent task space. However, even though the actual mapping from trajectories to tasks is unknown, it is possible to identify trajectories with similar latent behaviours. To construct similarity batches from the data, we considered that each location where a tourist stays for more than 2 minutes is a potential task ("attraction"). The 2 minutes threshold was chosen to reduce the number of irrelevant stops. In the $PK$ batches, the classes are $P$ locations (stops), sparsely sampled. For each location, $K$ trajectories heading to the same vicinity were sampled. On figure 3a, we plot the dataset of all the collected trajectories. The locations where the tourists stopped for more than 2 minutes are shown with circles. In figure 3b we plot an example of a $PK$ trajectory batch that we use for the optimisation. Training and implementation details are provided in Appendix B.

In this real life problem, trajectories are not annotated, so evaluating the separability of the tasks is difficult if not impossible. As an alternative, we propose to qualitatively evaluate the obtained representation function. Recall that our $PK$ batch procedure implied that the destination of the tourist is an indicator of the task it is accomplishing. A possible method to evaluate the performances of the obtained representation, is to sample trajectories that have similar representation and verify if these trajectories converge physically to a common area.
In Figure 4, trajectories with similar representations were illustrated in red. As expected, the obtained representation clusters trajectories with similar latent tasks (destinations). This confirms that the obtained representation succeeded in separating tasks and thus expanding the associated NM-RDP into an MDP. Moreover, as the transformation clusters trajectories according to their destination, we can potentially exploit that to analyse the tourists' patterns of visiting Salzburg.

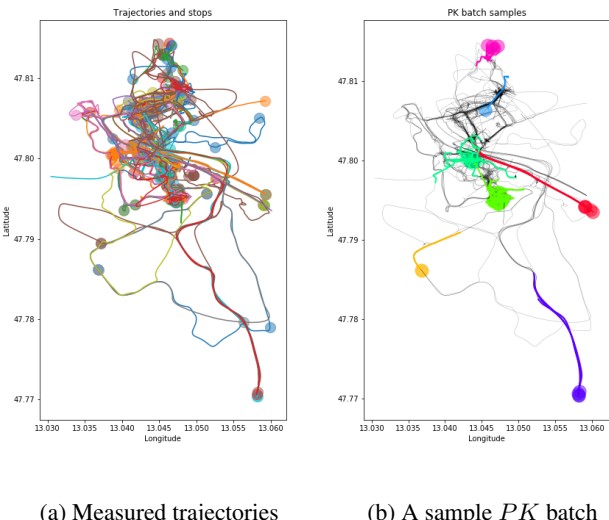

(a) Measured trajectories       (b) A sample $PK$ batch

Figure 3: Tourists GPS trajectory data

## 6 CONCLUSION

Expanding NMRDPs state spaces into minimal MDPs that share the same latent space structure is a common practice to identify optimal policies in those contexts. In contrastive learning, looking for separable representations with respect to a particular goal is also a common practice to simplify more complex tasks. The principal novelty of our work is leveraging representation learning techniques to bridge planning in non Markovian contexts with state of the art reinforcement learning techniques without the need for extensive knowledge about the NMRDP reward. Gathering a set of comparable

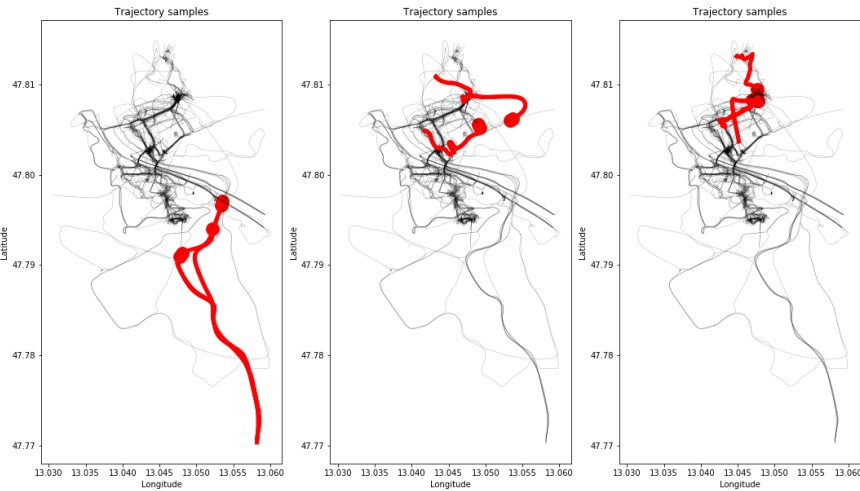

Figure 4: Sampled trajectories with similar representations

trajectories with respect to the associated latent task requires a weaker domain specific knowledge of the NMRDP problem than providing "relevant propositions" about the reward structure.

The fact that the obtained representation could be used to identify behavioural patterns of the tourists (agents) motivates our future work to consider inverse reinforcement learning in Non Markovian Decision Process.

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

## A    ANNEALING THE BATCH DIFFICULTY OVER TRAINING

Using hard samples provides a lot of nuanced information to the gradient computations which helps the generalisation property of the representation as it becomes harder to over fit. However, it is a recurrent issue (Hermans et al., 2017) that such sampling makes learning a lot harder in the beginning of the training. In fact the representation is still erratic and does not benefit much from difficult examples. We anneal the complexity of the batches as the representation is getting better at separating the tasks.

To circumvent this isue, we define a relaxed version of the batch hard loss defined in Equation 9 where we use the worst positive and negative sample per anchor with probability $p_{hard} \in [0, 1]$ and use a random positive and a random negative sample with probability $1 - p_{hard}$.

Formally, Let $P \sim \mathcal{B}(PK, p_{hard})$ a random variable of $PK$ binomial trials. We associate an anchor $\gamma_i^j$ to the trial $P_i^j$. A hard sample is used when $P_i^j = 1$ and a random sample is used when $P_i^j \neq 1$. The relaxed loss is defined bellow in Equations 12

$$\mathcal{L}_{BH}^{p_{hard}}(\theta, \gamma) = \sum_{j=1}^{P} \sum_{i=1}^{K} \log 1\mathrm{p} \left( m + ||\phi(\theta)_i^j - \phi(\theta)_{p(i,j)}^j|| - ||\phi(\theta)_i^j - \phi(\theta)_{n(i,j)}^{c(i,j)}|| \right), \qquad (12)$$

where

$$\begin{cases} p(i,j) := \begin{cases} \arg\max_{p \leq K} ||\phi(\theta)_i^j - \phi(\theta)_p^j|| & \text{if} \quad P_i^j = 1 \\ \sim \mathcal{U}_{\{1,..,K\}\setminus\{i\}} & \text{if} \quad P_i^j = 0 \end{cases} \\ n(i,j), c(i,j) := \begin{cases} \arg\min_{n \leq K, c \leq P, c \neq j} ||\phi(\theta)_i^j - \phi(\theta)_n^c|| & \text{if} \quad P_i^j = 1 \\ \sim \mathcal{U}_{\{1,..,K\}}, \mathcal{U}_{\{1,..,P\}\setminus\{j\}} & \text{if} \quad P_i^j = 0 \end{cases} \end{cases} \qquad (13)$$

It is clear that $\mathcal{L}_{BH}^{p_{hard}=1} = \mathcal{L}_{BH}$. However, as explained, optimising $\mathcal{L}_{BH}$ is difficult from a random initialisation. In what follows we develop formally an annealing procedure that increases $p_{hard}$ as the quality of the representation improves.

Let us denote by $\mathcal{P}_\theta^-$ and $\mathcal{P}_\theta^+$ the probability distributions of distances between the representations of negative and positive samples. Let $(L_\theta^+, (\sigma_\theta^+)^2)$ and $(L_\theta^-, (\sigma_\theta^-)^2)$ be the associated means and

variances. The average positive distance over a random batch of size $N = PK$, denoted by $\bar{l}_\theta^+(N)$, is an unbiased estimator of the mean as

$$\bar{l}_\theta^+(N) \sim \mathcal{N}(L_\theta^+, \frac{(\sigma_\theta^+)^2}{N}) \ , \ \bar{l}_\theta^-(N) \sim \mathcal{N}(L_\theta^-, \frac{(\sigma_\theta^-)^2}{N}).$$

As a consequence, the probability of sampling an average value bigger than $L_\theta^+ - 3\frac{\sigma_\theta^+}{\sqrt{N}}$ is roughly $0.9987 = 1 - 0.0013 \simeq \frac{1}{2}\mathbb{P}(\bar{l}^+(N) > L_\theta^+ - 3\frac{\sigma_\theta^+}{\sqrt{N}})$. Let us denote by $\bar{l}_b^+$ the average positive distance over the batch $b$. If batches were i.i.d. (which is not the case as $\theta$ evolves over time), and if we denote by $\min_{b \in [1,B]} \bar{l}_b^+$ the minimum average distance between negative samples over $B$ batches, the following would hold:

$$\mathbb{P}(\min_{b \in [1,B]} \bar{l}_b^+ > L_\theta^+ - 3\frac{\sigma_\theta^+}{\sqrt{N}}) \simeq 0.9987^B.$$

The same reasoning holds for $\max_{b \in [1,B]} \bar{l}_b^-$:

$$\mathbb{P}(\max_{b \in [1,B]} \bar{l}_b^- < L_\theta^- + 3\frac{\sigma_\theta^-}{\sqrt{N}}) \simeq 0.9987^B.$$

Consider the events $A^+ = \{\min_{b \in [1,B]} \bar{l}_b^+ < L_\theta^+ - 3\frac{\sigma_\theta^+}{\sqrt{N}}\}$ and $A^- = \{\max_{b \in [1,B]} \bar{l}_b^- > L_\theta^- + 3\frac{\sigma_\theta^+}{\sqrt{N}}\}$, the following property holds:

$$\mathbb{P}(A^+ \cup A^-) \simeq (1 - 0.9987^B)^2.$$

This leads to the following conclusion:

$$\mathbb{P}\left[L_\theta^+ - 3\frac{\sigma_\theta^+}{\sqrt{N}} > L_\theta^- + 3\frac{\sigma_\theta^+}{\sqrt{N}} \Big| \min_{b \in [1,B]} \bar{l}_b^+ > \max_{b \in [1,B]} \bar{l}_b^-\right] > (1 - 0.9987^B)^2. \quad (14)$$

For $B = 3000$, we have that $(1 - 0.9987^B)^2 \simeq 0.96$. We start with a low hard sampling probability and each time $\min_{b \in [1,B]} \bar{l}_b^+ > \max_{b \in [1,B]} \bar{l}_b^-$ we increase $p_{hard}$.

Figure 5 provides experimental analysis of the proposed procedure. In Figure 5a, we present the representation evaluation of the same problem addressed in Section 5.1. The obtained representation can be separated more easily. This allows us to conclude that obtained representation using the annealing procedure is a better approximation. The training details are provided in Figure 5b. From left to right we have illustrated the average loss value over the last used batches, the average distance between positive and negative samples as well as $\min_{b \in [1,B]} \bar{l}_b^+$ and $\max_{b \in [1,B]} \bar{l}^-$ and the value of $p_{hard}$ along the training. As the hard sampling probability increases, the loss value increases. This is an expected behaviours as we are measuring a more difficult version of the loss. Starting off with $p_{hard} = 1$ from a randomly initialised representation function did not converge in our toy example. In Section 5, we optimised $\mathcal{L}_{BH}^{p_{hard}=.1}$ to provide trajectory representation results.

## B  GPS TRACKS TRAINING DETAILS

The collected GPS tracks required prepossessing in order to be used for the trajectory representation learning. We constrained the observable space to a window around the city of Salzburg to avoid outliers in the data (such as tourists wondering off to surrounding cities, in Figure 6a, we observed that these trips render the data useless). The data kept for training are the GPS tracks with longitude within $[13.03, 13.06]$ and latitude within $[47.77, 47.82]$.

The obtained measurement are noisy. To smooth trajectories, we apply a Kalman filter. We model humans as a moving particle, characterised with $\mathbf{x} = [x, \dot{x}, y, \dot{y}]$, where $x$ and $y$ are the respective longitude and latitude. As we want to reduce the noise of the measured positions, we can define the observable as $\mathbf{z} = [x, y]$ and the observation matrix $\mathbf{H}$ as:

$$\mathbf{H} = \begin{pmatrix} 1 & 0 & 0 & 0 \\ 0 & 0 & 1 & 0 \end{pmatrix}. \quad (15)$$

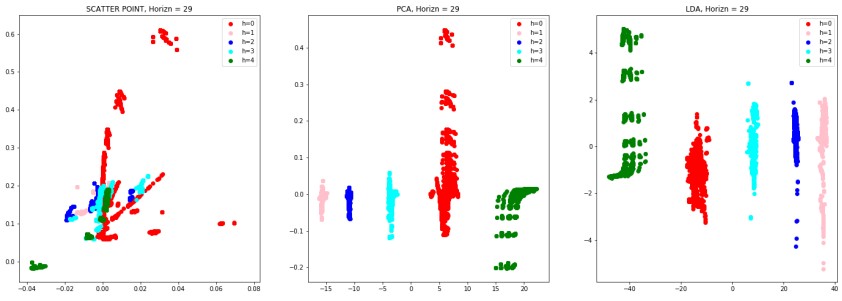

(a) Learned representation using annealed hard sampling probability

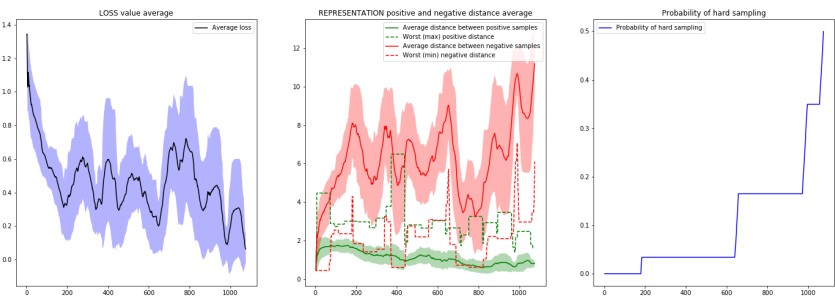

(b) Training evaluation of annealed hard sampling

Figure 5: Evaluating annealing

A linear approximation of the system dynamics is used to establish, for the $k^{th}$ observation, the following:

$$\begin{cases} x_k = x_{k-1} + \dot{x}_{k-1} \\ \dot{x}_k = \dot{x}_{k-1} \\ y_k = y_{k-1} + \dot{y}_{k-1} \\ \dot{y}_k = \dot{y}_{k-1} \end{cases}. \tag{16}$$

According to Equation 16, the transition matrix $\mathbf{T}$ is defined as follows:

$$\mathbf{T} = \begin{pmatrix} 1 & 1 & 0 & 0 \\ 0 & 1 & 0 & 0 \\ 0 & 0 & 1 & 1 \\ 0 & 0 & 0 & 1 \end{pmatrix}. \tag{17}$$

Using the observation matrix from Equation 15 and the transition matrix from Equation 17, we apply a Kalman filter to denoise the GPS tracks and smooth the trajectories. We used the implementation from the *python* library *pykalman*. In Figure 6, we illustrate the GPS tracks with and without the pre-processing operations. In Section 5.2, we use the data illustrated in Figure 6c.

### B.1 TRAINING HYPER-PARAMETERS

When constructing the trajectory batches, we sampled $P = 10$ stops that are at least $5 \times 10^{-3}$ far from each other[1]. For each stop, we sampled $K = 64$ trajectories heading, in the next 40 observations, to a point at worst $10^{-3}$ apart from the sampled stop. Figure **??** is a sample batch. We run 5000 iteration to optimise the relaxed batch hard loss $\mathcal{L}_{BH}^{Phard=.1}$, using the *Adam* routine from *TensorFlow* with a learning rate of $10^{-3}$, a momentum of .95 and a minimum gradient of $0.01$. In Figure 7, we illustrated the training results. The representation is indeed learning to separate trajectories that do not share the same destination. Therefore, we can conclude with more confidence that our representation separates the latent tasks of the agents (tourists).

---

[1]For the sake of simplicity, the distance is evaluated as if the GPS coordinates were in an Euclidean plan.

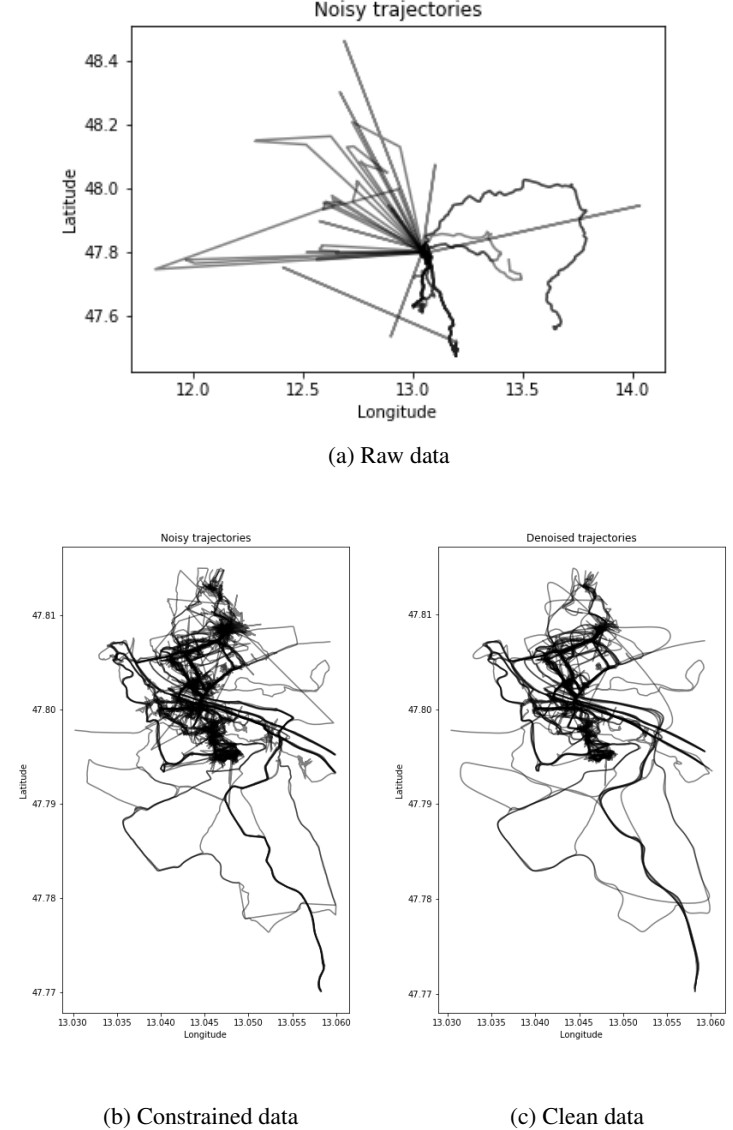

(a) Raw data

(b) Constrained data                    (c) Clean data

Figure 6: Pre-processing the data to smooth trajectories

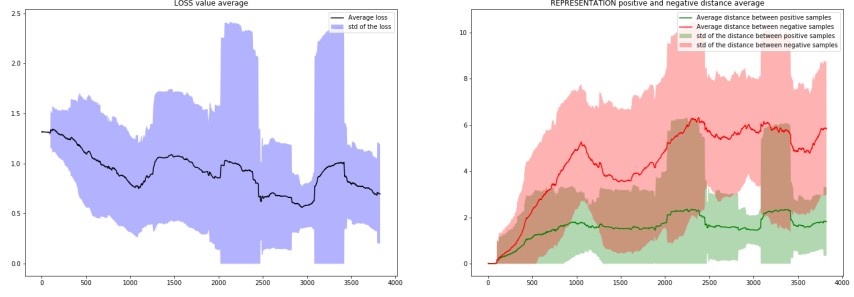

Figure 7: Optimising the trajectory representation

