# OpenReview forum: "Trajectory representation learning for Multi-Task NMRDPs planning"
_ICLR.cc/2020/Conference — Reject_

### Official Review · AnonReviewer1 · 2019-10-21
**Official Blind Review #1**

**Rating:** 6

**Review:**

Summary of the Paper:

        This paper proposes a method to address Non Markovian Reward Decision Processes using RL. For this, NMRPDs are transformed into Markovian Decision Processes. The idea is that the reward function can only depend on the last state of the trajectory and the task. A task representation is introduced. This is learned recursively. The proposed approach is evaluated in several experiments.

Detailed comments:

A point of criticism is that it seems that the authors do not compare with similar or related methods in the experiments. In any case, I think that the paper is interesting and will receive the attention of the community.

The paper is well written in general with only some typos: E.g.

"that separate"

"these proposition"


**Experience Assessment:**

I do not know much about this area.

**Review Assessment: Checking Correctness Of Derivations And Theory:**

I did not assess the derivations or theory.

**Review Assessment: Checking Correctness Of Experiments:**

I did not assess the experiments.

**Review Assessment: Thoroughness In Paper Reading:**

I made a quick assessment of this paper.

---

> ### Author Response · Authors · 2019-11-14
> **We thank the reviewer's comments and hope the following reply addresses their concerns.**
>
> Detailed comments:
> A point of criticism is that it seems that the authors do not compare with similar or related methods in the experiments. In any case, I think that the paper is interesting and will receive the attention of the community.
>
> > The main reason we do not compare to similar methods is that our work is the first attempt to expand NMRDPs in an unsupervised (weakly supervised at most) way using neural networks. However we agree that it’s important to quantify the usefulness of the obtained representation with respect to standard RL techniques. As such in the new uploaded version, we provide a comparison between
> a- an agent using DQN where the input contains the learned representations
> b- an agent using DQN but only access state features.
> c- agents using RDQN accessing state features with and without a history of the rewards
> The used environment and the learned representations are those presented in section 5.1.
> The agent using a trajectory representation outperforms all 3 other agents.
>
> >We also established that using representations lead to the actual optimal policies by evaluating the performances of a DQN-agent when using the learned representations compared to a one hot encoding of the tasks.
>
> The paper is well written in general with only some typos:
> E.g. "that separate" "these proposition"
> > We thank you for your appreciation. We took care of these typos in the new version.

---

### Official Review · AnonReviewer3 · 2019-10-22
**Official Blind Review #3**

**Rating:** 3

**Review:**

** Summary
The paper studies a specific class of Non-Markovian Reward Decision Processes (NMRDPs). In general, in NMRDP the dynamics is Markovian but the reward function may depend on the entire trajectory. The authors consider a sub-case where the trajectory can be mapped to a specific "task" and the reward function can be formalized as a mapping between the state-task pair to the reward. This greatly simplifies the problem that can be mapped onto an augmented MDP and solved using standard RL tools. The authors focus on the representation learning problem of the mapping between trajectories to an embedding of the task itself. In particular, they consider contrastive learning to find a representation that discriminate between trajectories associated to the same task and trajectories coming from different tasks. The resulting LSTM-based architecture is then evaluated on a grid-world synthetic problem and on GPS trajectories from tourists in the city of Salzburg.

** Overall evaluation
My main concerns about the paper is that the exact objective of the setting and the representation learning is not that clear and the empirical validation is limited to a qualitative assessment of the representation learning with no down-stream task. More in detail.

1- The assumption of task-dependent reward functions is very sensible. Yet I wonder to which extent it overlaps with literature in multi-task and meta-learning, or more in general with embedding of time-series (the trajectory in your case). It would be useful to frame/compare at the conceptual level the similarities and differences of the proposed setting with those fields.
2- At training time, the PK dataset is somehow supervised, as it is possible to know which trajectories are associated to the same task. At test time, the learned f_theta recursively maps trajectories to tasks. As such, it seems like it could effectively detect changes into the task itself by tracking how the trajectory evolves (when the task function is unknown). This aspect is never really evaluated in the empirical section, but it would be one of the most interesting uses of the learned representation.
3- If the task function T is known, it means that standard RL techniques can be used to solve MDP N. The actual advantage of using a specific function to embed the task to a space in Re^d is never really explained in the paper. Does it make solving N simpler or more effective than using a simple encoding for the task? No evaluation is available in this sense.
4- The empirical evaluation is limited to qualitative analysis of the representation learned in the problem. Yet, there is no clear support that the representation is good/useful in a more quantitative way (e.g., by actually solving the RL part or in identifying quickly the current task).

Some more specific questions/comments

1- Some of the notation is a bit redundant. For instance, phi is mapping from H to Re^d, while f is a mapping from trajectories to Re^d, but in the end they are doing the same thing, as H itself can be obtained as a mapping from trajectories to tasks through T.
2- "Using RL techniques, finding an optimal policy pi* in the equivalent MDP \hat M is possible and by extension pi* is optimal for N". This passage is not fully clear, if phi is introducing some form of approximation, then pi* may no longer be optimal for the original MDP N. On the other hand, if phi is not approximating but "just" changing the representation from H to Re^d, then it is not clear what is the interest of it.
3- The assumption that similar trajectories can be identified is somehow strong. It would be good to have a more thorough support for it.
4- It is not fully clear what L_BH^local is indeed a local loss. It seems like you are simply using the mapping from the whole trajectory. Is this why it is called local?
5- While I appreciate that the introduction is sketching many different scenarios to support the models studied in the paper, in the end they mostly lack of depth and they rather give a confusing impression instead of clarifying in a compelling way what is the problem studied in the paper. I suggest you rather pick one single scenario with a good level of detail to provide a more solid support to the paper.

** Minor comments
1- You often use "he" to refer to the agent. It would be better to use "it" or "she".

**Experience Assessment:**

I have published in this field for several years.

**Review Assessment: Checking Correctness Of Derivations And Theory:**

I carefully checked the derivations and theory.

**Review Assessment: Checking Correctness Of Experiments:**

I assessed the sensibility of the experiments.

**Review Assessment: Thoroughness In Paper Reading:**

I read the paper thoroughly.

---

> ### Author Response · Authors · 2019-11-14
> **We thank the reviewers comments, We provide quantitative proof that the proposed approach works**
>
> Overall evaluation
> My main concerns about the paper [..] with no down-stream task.
> > We have updated the introduction that now ends with a clear summary of our contributions and some highlights of our objectives. We also include an evaluation of a DQN agent performance in the down-stream task introduced in section 5.1 (grid world)
>
> 1- The assumption of task-dependent reward functions is very sensible. [..]
>
> > To the extent of our knowledge, meta-learning and multi-task reinforcement learning investigates the ability to identify optimal policies for similar problems. However they satisfy the markovian property by including the task nature in the state space (for example if the task is to teach an agent to go to multiple grid cells, the state he would use contains the current position and the target position).
> Even though they do not explicitly say that the reward is indeed task-dependent,  successfully using the bellman equation to solve this kind of settings implies that the reward is markovian with respect to the chosen state, and thus with respect to the task.
> We precise in the introduction of the new version that we can exploit recent work in those fields to learn optimal policies once the trajectory representation function is learned.
>
> 2- At training time, the PK data set is somehow supervised, [..] the most interesting uses of the learned representation.
>
> >Our approach is indeed not fully unsupervised, but “weakly” supervised. However providing a PK dataset does not require knowing the T function. As illustrated in the examples given in the introduction, there are many cases where the construction of these batches is possible without accessing the T function (e.g. in the waiter example, trajectory similarity can be induced from a similarity in ratings provided it’s the same client). In the experimental setting, we do not know the function that maps trajectories to tasks, however we can identify trajectories that headed to the same destination.
> > Although the obtained representation function could be used to reverse engineer the T function, this is not necessary in our context as the overarching objective is to identify an equivalent MDP to the considered NMRDP.
>
> 3- If the task function T is known, [..] No evaluation is available in this sense.
>
> > It is true that in synthetic scenarios, the actual T function is available. In such experimental setting, as the one we propose in section 5.1 (the grid-world NMRDP), we can provide the comparison proposed by the reviewer. This will indeed confirm to what extent the obtained representations are useful to approximate an equivalent MDP. We provide in the updated version, a comparison between a DQN using the learned representations and a DQN agent using a one hot encoding of the actual current task.
> > As expected, we obtained comparable performances for the two agents, with faster convergence in the case we use the one hot encoding.
>
> 4- The empirical evaluation [..] (e.g., by actually solving the RL part or in identifying quickly the current task).
>
> > We agree that solving a down-stream task is relevant to the evaluation of our representation. In the new uploaded version, we provide a comparison between
> a- an agent using DQN where the input contain the learned representations
> b- an agent using DQN but only access state features.
> c- agents using RDQN accessing state features with and without a history of the rewards
> The used environment and the learned representations are those presented in section 5.1.
> The agent using a trajectory representation outperforms all 3 other agents.

---

> ### Author Response · Authors · 2019-11-14
> **Reply to the specific questions/comments**
>
> Some more specific questions/comments
> 1- Some of the notation is a bit redundant. [..]
> > In the considered class of problems, \phi is an unknown feature function that can be used to construct an equivalent MDP. For any injective feature function we have a candidate “ideal” trajectory representation function \phi o T : \Gamma \rightarrow Re^d.
> On the other hand, f_\theta is an approximation of such functions.
> We revisited them in the new version. \phi now always denote trajectory representations. f_\theta is only used for the approximation. We define a classifier C: Re^d \rightarrow H to have a coherent formulation.
>
> 2- "Using RL techniques,[..] is optimal for N". This passage is not fully clear [..]
> > We realise that our wording might have been misleading in this sense. \hat(M) does not introduce an approximation of the equivalent MDP. The reason we introduce \phi is that most RL techniques use features. We wanted our approach to be adaptable to all down-stream RL algorithm and thus provided of a feature based equivalent MDP template for our approximation. We modified section 3.1 to provide a simpler, and less constraining formulation of such equivalent MDPs.
>
> 3- The assumption that similar trajectories can be identified is somehow strong. It would be good to have a more thorough support for it.
> > We provide multiple examples in the introduction that support this hypothesis, namely the waiter problem. The rating of a client can be used as a weak supervision to construct the similarity batches.
> Furthermore, in the Salzburg city data set, we do not possess a signal that identifies the tasks. However we were able to construct a batch of similar behaviors based on the place where the tourists went.
> > We also want to highlight, that our method, requires a weaker assumption that standard techniques in expanding NMRDPs into MDPs. In other words, all previous works that tackle the expansion problem require stronger assumptions.
>
> 4- It is not fully clear what L_BH^local is indeed a local loss. It seems like you are simply using the mapping from the whole trajectory. Is this why it is called local?
> > the loss is denoted as “local” because we are only propagating the gradient to the current time step, more or less as if the previous representation is correct. By alternating the minimization of this loss and the re-evaluation of the representations, we can optimize the initial loss. We reworked section 4.2 to make the proposed algorithm clearer.
>
> 5- While I appreciate that the introduction is sketching many different scenarios [..] I suggest you rather pick one single scenario with a good level of detail to provide a more solid support to the paper.
> > The reviewer's question #3 is what motivated us to provide multiple examples. We understand that our work does not follow the usual frameworks of NMRDP expansion and wanted to provide multiple scenarios where this approach is applicable.
>
> ** Minor comments
> 1- You often use "he" to refer to the agent. It would be better to use "it" or "she".
> >Thanks for noting it, we will change them.

---

### Official Review · AnonReviewer2 · 2019-11-03
**Official Blind Review #2**

**Rating:** 3

**Review:**

Short summary of paper:
The paper investigates representation learning  from trajectories in a framework which generalizes MDPs (NMRDPs), in which rewards are non-Markovian and follow a hidden process (this can be seen as a more structured, special case of a POMDP). The authors suggest learning a trajectory embedding by using a triplet loss, justified by a sufficient condition for learning an embedding which corresponds to the true task id.

Overall I felt the paper fell a bit short of the standards for ICLR, based on subpar writing, lack of comparison to convincing baselines, and unclear applicability to more complex environments.

Main issues:
- The methodological sections of the paper seems conceptually accessible while being written in an overly mathematical, meandering fashion, using heavy, sometimes unclear notation, or referring to notation from other papers.

As an example, \mathcal{M}^* is first introduced by reference to another paper (the notation does not really need to be introduced until the mapping is made precise later in the paper); \phi(\theta)_i^j(-1) is an unnecessarily heavy, unpleasant notation for the concept used.
It was unclear if T is meant to represent the total trajectory length or a time index of a given trajectory (the notation would imply the former, the way it used, the latter).  Similarly equation (6) is a very mathematical way to denote a relatively simple condition (which could be written more simply by comparing inf and sup distances between two trajectory embeddings corresponding to different or identical latent h). Again, equation (7) seems like a contrived way to write a relatively simple concept, and the simple proof that (6) implies (7) is not included.
I feel this section could be improved by using precise but as simple notation as possible, and a sequence of propositions explaining why the triplet loss allows for identifiability of the reward process, with a clear flow between statements.

- Lack of baselines:
No alternative or baseline seems considered in the paper. The setting is perhaps a bit unusual, but since the data considered is essentially sequence of observations, any model of sequences could be used to produce embeddings (for instance, recurrent VAE, autoregressive models, time-contrastive methods) and compare the different trajectory representations.

- Unclear generalization to more complex problems:
(6) only implies (7) as far as I can tell, is sufficient but not necessary, and may be too strong of a condition to enforce; similary, theoretically requiring injectivity of phi seems practically too strong of a condition, as it essentially requires hashing each trajectory into a different embedding, in spite of the fact that many aspects of observations may be irrelevant for the task at hand). A key difficulty to scale this algorithm will be to find how to distance between trajectories in high dimension - perhaps the most important question for representation learning of trajectories, and a point the paper does not address.


Positives:
The method is overall well principled and the theoretical justification of the triplet loss is interesting. The dataset used for experiments is also interesting.

Minor/Questions:
- IIUC, the equation (3) needs to hold for all times T, this needs to be clearly stated as often T is used to denote total episode length.
- Bottom of page 4, 'The construction of the task ... \phi_T=\phi(\mathcal T(o_T, \phi_{T-1}); does this not require injectivity of \phi, which is only stated afterwards?
- Calling the algorithm 'EM' is a bit of a stretch (there are no latent variables), it is better to call it a form of coordinate ascent.
- The way the EM-like algorithm is set up, it appears the gradient with respect to theta is only computed with respect to a single time step (similar to elman networks style truncated backprop) - why not use regular backpropagation through time to train though the entire sequence?
- typo on section 5.1: 'Grid-world problem'
- It is very hard to see on figure 3 what we are supposed to see (I don't see these as having the same destination, perhaps highlight more clearly?)


**Experience Assessment:**

I have published in this field for several years.

**Review Assessment: Checking Correctness Of Derivations And Theory:**

I assessed the sensibility of the derivations and theory.

**Review Assessment: Checking Correctness Of Experiments:**

I assessed the sensibility of the experiments.

**Review Assessment: Thoroughness In Paper Reading:**

I read the paper at least twice and used my best judgement in assessing the paper.

---

> ### Author Response · Authors · 2019-11-14
> **We thank the reviewer's comments and hope the following reply addresses their concerns.**
>
> Main issues:
> - The methodological sections of the paper [...] with a clear flow between statements.
>
> > We took note of the provided methodological remarks. In the new version of the paper, we precise when the equation hold for any horizon T to ease the distinction. We alleviated some of the notations as mentioned, re-formulated Section 4 and provided a constructive proof of how (6) implies (7).
>
> - Lack of baselines: [...] compare the different trajectory representations.
>
> > Given that our work is a first attempt in “neutralizing” the problem of expanding NMRDPs into MDPs, and given that we are also the first attempt to solve the problem in a weakly supervised framework, comparing our work to alternative/off-the-shelf baselines can not be done in an experiment that would be “fair” towards competitors which are not designed for the same objectives.
> We also believe that experimentally trying multiple sequence embedding models (and quantifying their performances) is beyond the scope of this paper – actually, we tried several of them before choosing the final one. It is clear that different embeddings would also work, with certainly similar properties (maybe slightly better, probably slightly worse). But the scope of the paper was to justify the overall method, rather than fine tuning and optimizing it (and we are fairly confident that the “best” embedding actually depends on the problem at hand) (note that this is actually ongoing work)
> However we agree that providing quantitative performance of the model is an interesting contribution to the paper. In the revised and uploaded version, we provide a comparison between:
> a- an agent using DQN where the input contain the learned representations
> b- an agent using DQN but only access state features.
> c- agents using RDQN accessing state features with and without a history of the rewards
> The used environment and the learned representations are those presented in section 5.1.
> The agent using a trajectory representation outperforms all 3 other agents.
>
> - Unclear generalization to more complex problems: [..] a point the paper does not address.
>
> > \phi is a feature function introduced to construct an equivalent MDP using the latent tasks.
> \phi is a function from H to R^d, T is a function from \Gamma to H. \phi being injective does not imply that \phi \circ T is injective. And thus we do not require to hash all trajectories into different features.
> \phi’s injectifity (between H and R^d) ensure the equivalence between \tilde\mathcal{M} and \mathcal{M}^* and by transition \mathcal{N}.
> > Morally, in our approach we approximate the injectifity property of \phi with the separability of trajectory representations with respect to tasks (i.e. eq (6)).
> However it’s true that this remain a strong assumption. In the new version, we modify section 3.1, to provide a more relaxed condition of \phi. We define \phi now as a function from trajectories to R^d, and only require the existence of a classifier C such that C o \phi = T.
> We also precise in the new version that f_\theta can be used either to compute the tasks or to aproximate \hat{\mathcal{M}} by substituting \phi with f_\theta.
> > We agree that the curse of dimensionality is one of the main issues in trajectory representation. We consider that treating this problem is beyond the scope of this paper as our proposition to use trajectory representation learning to expand NMRDPs into MDPs can eventually include breakthroughs that solve trajectory representation in the high dimension case. However we argue that not all real life examples are high dimensional and our approach can be used as is.
>
> Positives:
> The method is overall well principled and the theoretical justification of the triplet loss is interesting. The dataset used for experiments is also interesting.
>
> > We thank you for your appreciation. We highlight that the dataset used in the experiment encouraged us to pursue the IRL problem in NMRDPs using the same principles introduced in this paper.
>
> Minor/Questions:
> - Bottom of page 4 [..] which is only stated afterwards?
> > Eq (5) introduces a system that need to be solved to find \mathcal{M}^*. given an injective function \phi, we can construct a solution. We are providing a particular solution to the problem and we did not solve for all possible expansion of the NMRDP.
> Section 3.1 is modified as explained, however the reasoning is the same, the required condition is sufficient but not required to solve Eq (5), this is why we state it afterwards.
>
> - Calling the algorithm 'EM' is a bit of a stretch [ .. ]  why not use regular backpropagation through time to train though the entire sequence?
> > We agree that TBPTT is a more suitable framework to describe our approach. However we constructed it in a similar fashion to EM. We changed the way we introduced this in the new version of the paper.
> What lead us to such choice is the empirical stability and speed of the proposed approach.

---

### Decision · Program_Chairs · 2019-12-19

**Decision:**

Reject

**Comment:**

The paper considers a special case of decision making processes with
non-Markovian reward functions, where conditioned on an unobserved task-label
the reward function becomes Markovian.
A semi-supervised loss for learning trajectory embeddings is proposed.
The approach is tested on a multi-task grid-world environment and ablation
studies are performed.

The reviewers mainly criticize the experiments in the paper. The environments
studied are quite simple, leaving it uncertain if the approach still works in
more complex settings.
Apart from ablation results, no baselines were presented although the setting is
similar to continual learning / multi-task learning (with unobserved task label)
where prior work does exist.
Furthermore, the writing was found to be partially lacking in clarity, although
the authors addressed this in the rebuttal.

The paper is somewhat below acceptance threshold, judging from reviews and my own
reading, mostly due to lack of convincing experiments. Furthermore, the general setting
considered in this paper seems quite specific, and therefore of limited impact.